# Tumor-Targeted Delivery of the p53-Activating Peptide VIP116 with PEG-Stabilized Lipodisks

**DOI:** 10.3390/nano10040783

**Published:** 2020-04-19

**Authors:** Sara Lundsten, Víctor Agmo Hernández, Lars Gedda, Tina Sarén, Christopher J. Brown, David P. Lane, Katarina Edwards, Marika Nestor

**Affiliations:** 1Department of Immunology, Genetics and Pathology, Uppsala University, 751 85 Uppsala, Sweden; sara.lundsten@igp.uu.se (S.L.); tina.saren@igp.uu.se (T.S.); 2Department of Chemistry—BMC, Uppsala University, 751 24 Uppsala, Sweden; victor.agmo@kemi.uu.se (V.A.H.); lars.gedda@kemi.uu.se (L.G.); katarina.edwards@kemi.uu.se (K.E.); 3Department of Pharmacy, Uppsala University, 751 23 Uppsala, Sweden; 4p53 lab, Agency for Science, Technology and Research (A*STAR), Singapore 138648, Singapore; cjbrown@p53lab.a-star.edu.sg (C.J.B.); dplane@p53lab.a-star.edu.sg (D.P.L.); 5Department of Microbiology, Tumor and Cell Biology, Science for Life Laboratory, Karolinska Institutet, 171 65 Solna, Stockholm, Sweden

**Keywords:** targeted tumor therapy, targeted drug delivery, Lipid bilayer disks, lipodisks, nanocarriers, p53, stapled peptides

## Abstract

Stapled peptides targeting the interaction between p53 and its negative regulators MDM2 and MDM4 have exhibited great potential as anti-cancer drugs, albeit with room for improvement in formulation and tumor specificity. Lipid bilayer disks (lipodisks) have emerged as promising drug nanocarriers and can by attachment of targeting moieties be directed selectively towards tumor cells. Tumor-targeted delivery of stapled peptides by use of lipodisks may therefore increase the uptake in the tumors and limit toxicity in healthy tissue. Here, we utilized epidermal growth factor receptor (EGFR)-targeted lipodisks to deliver p53-activating stapled peptide VIP116 to EGFR-expressing tumor cells. We demonstrate that VIP116 can be stably formulated in lipodisks (maximum peptide/lipid molar ratio 0.11). In vitro cell studies verify specific binding of EGF-decorated lipodisks to tumor cells and confirm that targeted delivery of VIP116 significantly decreases tumor cell viability.

## 1. Introduction

Over the past decades, cytotoxic chemotherapeutic drugs have played an important role in cancer therapy [1]. Regrettably, the efficacy is often limited by toxicity and resistance development [1]. As a greater understanding of the molecular landscape of cancer has emerged, treatments customized to the patients’ genetic and environmental background are being developed [2].

Tumor suppressors are central players in tumorigenesis. One well-known tumor suppressors is p53, a transcription factor often referred to as the guardian of the genome due to its role in sensing and responding to DNA damage [3]. Nearly all tumors have an inactive p53 pathway, e.g., due to mutation in the *p53* gene itself or repression of cellular p53 levels [4]. In case of wild type p53, overexpression of the negative regulator MDM2, and its structural homologue MDM4, is one important approach used by the tumor cells to keep p53 levels to a minimum (Figure 1) [3].

Reactivation of wild type p53 is considered an attractive method for cancer therapy. Thus, molecules that inhibit the p53-MDM2 interaction have been developed and several of these are in clinical trials [5]. However, the therapeutic response has been meager, which is attributed to short biological half-lives and hematological toxicities of the inhibitors, as well as to resistance caused by increased MDM4 activity [6,7].

VIP116 and its predecessor PM2 are stapled peptides that target both the p53-MDM2 and -MDM4 interactions [8,9]. Moreover, the stapling improves the in vivo half-life of the peptides [10]. We have previously demonstrated promising therapeutic effects in vivo of PM2 in wild type p53 cancer [11]. The peptides’ in vivo application can however be limited by e.g., low aqueous solubility or off target binding, and the effectiveness could be further improved by increased tumor targeting. These issues could potentially be overcome by formulating the peptides in tumor-targeted nanocarriers.

Lipid bilayer disks (lipodisks) are nanosized bilayer structures, stabilized into flat, circular shapes by polyethylene glycol (PEG)-linked lipids [12,13,14]. These nanoparticles show great potential as drug carriers and have been preclinically assessed for delivery of anti-cancer and anti-bacterial compounds [13,15,16,17,18,19]. Moreover, a targeting moiety can be attached to the lipodisk to increase delivery to desired tissues. Epidermal growth factor receptor (EGFR) is upregulated in several cancers, and has emerged as a target for diagnostic imaging and therapy [20]. Consequently, we have developed and utilized EGFR-targeted lipodisks for delivery of different classes of anti-cancer drugs [15,19]. In the present study, we investigated the feasibility of delivering the novel p53-activating peptide VIP116 to tumor cells via EGFR-targeted lipodisks.

## 2. Materials and Methods 

### 2.1. Production of Lipodisks and Liposomes

Dry 1,2-dipalmitoyl-sn-glycero-3-phosphocoline (DPPC) powder, 1,2-distearoyl-sn-glycero-3- phosphoethanolamine-N-(polyethylene glycol)-2000 (DSPE-PEG2000) and DSPE-PEG2000-biotin were purchased from Avanti Polar Lipids (Alabaster, AL, USA) or kindly gifted by Lipoid (Ludwigshafen, Germany). 1,2-Distearoyl-sn-glycero-3-phosphocholine (DSPC) was also gifted by Lipoid. DSPE-PEG3400-NHS was purchased from Shearwater Polymers (Huntsville, AL, USA).

Liposomes to be used for preparations of supported bilayers for QCM-D were composed of DPPC:DSPE-PEG2000 96:4 molar ratio. A lipid film was first prepared by dissolving the lipids in CHCl_3_ and dried under a stream of nitrogen gas. Remaining solvent was removed in vacuum overnight. The film was hydrated in phosphorous buffered saline (PBS) pH 7.4 at 60 °C for 30 min and subsequently freeze-thawed in 60 °C/liquid nitrogen and extruded at 60 °C through a 0.1 μm membrane (Whatman, GE Healthcare Bio-Sciences, Pittsburgh, PA, USA)

All lipodisks used in the study were produced with a method based on detergent depletion using Bio-Beads (SM-2 Adsorbent, Bio-rad, Sundbyberg, Sweden) in accordance with a previously described protocol [15,19]. Briefly, for lipodisks used in QCM-D assays a lipid film composed of DSPC:DSPE-PEG2000:DSPE-PEG2000-biotin 80:16:4 was prepared as described above. The film was hydrated in 31.5 mM Octylglucoside in HEPES buffered saline (HBS) at 60 °C for 30 min and subsequently incubated for 2 h with biobeads. The lipodisk solution was separated from biobeads with a 30G syringe.

For cellular assays, non-targeting and targeting lipodisks were prepared with DSPC:DSPE-PEG2000:DSPE-PEG3400-NHS 8:2:1. A lipid film with DSPC and DSPE-PEG2000 was prepared as described above. The lipid film and DSPE-PEG3400 were first hydrated separately in HBS with 41.5 mM octylglucoside in 60 °C for 30 min, then mixed and incubated for an additional 30 min. The solution was incubated with biobeads as described above. Excess octylglucoside was removed by use of spin column (Pierce Protein Concentrator, Thermo Fisher, Waltham, MA, USA). For EGFR-targeted lipodisks, 3.6 mg DSPE-PEG3400-NHS was instead conjugated to 300 μg murine EGF (EA140, Merck, Darmstadt, Germany) in PBS pH 7.4 overnight in room temperature under stirring. EGF-micelles was separated from unconjugated EGF using size exclusion chromatography (Sephadex G-150, Amersham Biosciences, Uppsala, Sweden) and thereafter added to the lipid film, as described above. Targeting lipodisks contained EGF with an approximate EGF/lipid molar ratio of 1.11 × 10^−3^ [15,19].

### 2.2. Cryo-TEM

The presence of lipodisks was verified with cryo-transmission electron microscopy (cryo-TEM) using a Zeiss Libra 120 Transmission Electron Microscope (Carl Zeiss NTS, Oberkochen, Germany) according to previously described protocol [21] Appendix A. The samples were equilibrated at 25 °C and high relative humidity (>90%) within a climate chamber, prior to vitrification. The microscope was operating at 80 kV and in zero loss bright-field mode. Digital images were recorded with a BioVision Pro-SM Slow Scan CCD camera (Proscan GmbH, Scheuring, Germany) and iTEM software (Olympus Soft Imaging System GmbH, Münster, Germany).

### 2.3. QCM-D

A quartz crystal microbalance with dissipation monitoring (QCM-D) was used to study the interaction of VIP116 with lipodisks and lipid membrane. Silica or biotynilated gold sensors (Q-Sense, Gothenburg, Sweden) was mounted on a QCM-D E1 (Q-sense) instrument thermostatted at 21 °C. Frequency and dissipation data were collected at the 3rd, 5th, 7th, 9th, 11th and 13th overtones and the immobilized mass was calculated as previously described [22]. Matlab R2016b (The MathWorks, Natick, MA, USA) was used for performing the calculations. A constant flow rate of 100 μL/min was used throughout the interaction assays.

For immobilized lipodisks, biotinylated gold sensors were first coated with streptavidin and subsequently biotinylated lipodisks. The system was then equilibrated with 0.5% DMSO in PBS pH 7.4 which was used throughout the assay. VIP116 with concentrations of 10, 30 and 100 nM was stepwise added. After the association phase, 0.5% DMSO in PBS was applied to study dissociation. To verify that VIP116 did not unspecifically bind to the bare sensor, a control experiment was performed without lipodisks (Appendix A). Kinetics constants were calculated by fitting the real-time R_eff_ data to a Langmuir 1:1 binding model with TraceDrawer (Ridgeview Instruments, Vänge, Sweden).

For the supported lipid bilayer in gel phase, silica sensors were first cleaned according to the instructions of the provider. (10 min UV/ozone treatment followed by immersion in SDS 2% and finally 10 min UV/ozone treatment). The sensor was then mounted on the instrument and exposed to a continuous flow of PEGylated DPPC liposomes in PBS (~1 mg/mL) until the surface was saturated. The system was rinsed with deaerated PBS, before the flow was stopped and a temperature program (a constant increase from 21 °C to 60 °C for 70 min, a temperature plateau of 60 °C for 25 min and a decrease back to 21 °C for 70 min) was applied and looped overnight to cause the liposomes to collapse and form a lipid bilayer in gel phase as confirmed by the final frequency change (−33 Hz) value and a very low dissipation factor (<4 × 10^−6^). After formation of the bilayer, the binding assay was performed with stepwise addition of 10, 30 and 100 nM VIP116. The R_eff_ at equilibrium for each concentration (average of last 100 data points before addition of next concentration or start of dissociation) for the VIP116-lipid bilayer interaction was compared to that of the VIP116-lipodisk interaction. For the VIP116-lipodisk interaction, the data was fitted to a Langmuir 1:1 model (one site—specific binding) to calculate kinetic parameters.

### 2.4. ^125^I-Labeling

Anti-EGFR antibody cetuximab (Merck), EGF and VIP116 were labeled with ^125^I according to the chloramine-T method, where protein and ^125^I are incubated with chloramine-T for 60 s on ice. The reaction was quenched by adding sodium metabisulfite. For cetuximab labeling, 0.4 nmol protein and 8 MBq ^125^I was labeled in PBS pH 7.4 using 10 μL 1 mg/mL chloramine-T and 20 μL 1 mg/mL sodium metabisulfite (yield 81–88%). For EGF, 1.7 nmol protein and 10 MBq ^125^I was labeled in PBS using 4 μL 1 mg/mL chloramine-T and 10 μL 1 mg/mL sodium metabisulfite (yield 83–89%). For VIP116, 40 nmol peptide was labeled in MilliQ water with 4 MBq ^125^I using 32 μL 2 mg/mL chloramine-T and 64 μL sodium metabisulfite (yield 52–54%). VIP116, ^125^I and NaI (VIP116:NaI molar ratio 1:1) was incubated for 20 min at room temperature prior to adding chloramine-T.

Labeling yield was assessed with instant thin layer chromatography (Biodex Medical Systems, New York, NY, USA) using 70% acetone (cetuximab) or 0.9% NaCl (VIP116 and EGF) as mobile phase. Chromatography stripes were analyzed with a phosphorImager (BAS-1800II, Fujifilm, Tokyo, Japan). Free ^125^I was removed by size exclusion chromatography, either NAP-5 (GE Healthcare) with PBS pH 7.4 as mobile phase (cetuximab, EGF) or PD MiniTrap G10 (GE Healthcare) with ethanol as mobile phase (VIP116).

### 2.5. Cell Lines and VIP116

HEK-293T (ATCC) cells was cultured in DMEM supplemented with 10% FBS, 1% Pest, 1% puromycin and 500 μg/mL geneticin. HCT116 (Horizon Discovery, Waterbeach, UK) cells were cultured in RPMI-1640 (Biochrom, Berlin, Germany) supplemented with 10% FBS (Sigma-Aldrich, Darmstadt, Germany), 1% Pest and 1% L-glutamine (Biochrom). VIP116 (late isomer) was produced by the p53 lab and a 10 mM stock solution in DMSO was prepared and kept at −20 °C [8,9].

### 2.6. EGFR-Transduction of HCT-116

A high-EGFR expressing cell line was created from HCT-116 cells with the use of lentiviral transduction. In brief, EGFR transcript variant 1 (Clone ID: OHu102757 Genescript, Piscataway, NJ, USA) was cloned into a third generation self-inactivating (SIN) lentiviral vector (SBI, System Biosciences, Mountain View, CA, CA), containing puromycin, under the control of elongation factor-1 alpha (EF1a) promoter (pBMN(EF1a-EGFRv1-Puro)). The sequence was purchased and cloned into the vector by Genescript. Vesicular stomatitis virus (VSV)-G pseudotyped lentiviral particles were produced in HEK-293T cells using polyethyleneimine (Sigma-Aldrich, St. Louis, MO, USA). Cells were infected with pBMN(EF1a-EGFRv1-Puro) and the pLP1, pLP2 and pLP/VSVG (Invitrogen, Carlsbad, CA, USA) packaging plasmids at a 2:1:1:1 ratio and viral supernatant was collected 48h post transfection. Viral supernatant was filtered and added to HCT116 cells. After 24 h RPMI-1640 medium supplemented with 2 μg/mL puromycin was added for selection. The new cell line will hereon be referred to as HCT-EGFR and was cultured in the same conditions as the parental cell line.

### 2.7. Verification of EGFR Expression

Cellular EGFR expression was confirmed by cellular uptake measurements of ^125^I-labeled anti-EGFR antibody cetuximab. HCT-116 or HCT-EGFR cells were seeded in 48 well plates (4 × 10^4^ cells per well) and incubated for 24 h. ^125^I-cetuximab was added to cells with concentrations ranging from 0.1–30 nM and cells were incubated at 37 °C for 4 h. To measure unspecific binding, additional wells were incubated with 30 nM ^125^I-cetuximab and a 100-fold excess unlabeled cetuximab. Cells were washed, trypsinized and counted. Cell-associated radioactivity was measure in a gamma counter (1480 Wizard 3”, Wallace, Turku, Finland). Data was fitted to a Langmuir 1:1 model (one site –specific binding) to calculate B_max_.

### 2.8. HCT-116 Western Blot Analysis

HCT-116 cells were seeded into 96 well plates at a cell density of 60,000 cells per well and incubated overnight. Cells were also maintained in DMEM cell media with 10% fetal bovine serum (FBS) and penicillin/streptomycin. Cell media was then removed and replaced with cell media containing tVIP116 and vehicle controls at the concentrations indicated in DMEM cell media with 2% FCS. After the stated incubation time (24 h) cells were rinsed with PBS and then harvested in 100 µl of 1 × NuPAGE LDS sample buffer supplied by Invitrogen (NP0008). Samples were then sonicated, heated to 90 °C for 5 min, sonicated twice for 10 s and centrifuged at 13, 000 rpm for 5 min. Protein concentrations were measured by BCA assay (Pierce BCA Protein assay kit, Thermo Fisher, Waltham, MA, USA). Samples were resolved on Tris-Glycine 4–20% gradient gels (Bio-rad, Sundbyberg, Sweden) according to the manufacturer’s protocol. Western transfer was performed with an Immuno-blot PVDF membrane (Bio-rad) using a Trans-Blot Turbo system (Bio-rad). Western blot staining was then performed using antibodies against actin (AC-15, Sigma) as a loading control, p21 (118 mouse monoclonal), Mdm2 (4B2 mouse monoclonal) and p53 (DO-1 mouse monoclonal).

### 2.9. p53 Gene Reporter Assay:

ARN8 cells stably expressing the p53 reporter RGCΔFos-LacZ were described previously [23,24,25]. ARN8 cells were maintained in Dulbecco’s Minimal Eagle Medium (DMEM) with 10% fetal calf serum (FCS) and 1% (*v/v*) penicillin/streptomycin. The *p53* gene reporter assay was performed as previously reported [23,24,25,26]. ARN8 cells were seeded into a 96-well plate at a cell density of 20,000 cells per well in DMEM containing 10% FCS and 1% penicillin/streptomycin and incubated for 24 h at 37 °C 5% CO_2_. Cell media was then removed and replaced with 100 μL of fresh DMEM containing 1% FCS and DMSO (0.1% *v/v*), Nutlin 3a (5 µM) or VIP116 (5, 10 or 20 µM) in quadruplicate wells. After 18 h, cells were lysed, and β-galactosidase activity determined using the β-galactosidase substrate CPRG. Wells containing no cells were included in the analysis to establish background. Quantification of the enzymatic product chlorophenol red was performed by measuring the absorbance at 595 nm using a Cytation5 plate reader (BioTek, Winooski, VT, USA), and statistical analysis was performed using GraphPad Prism 8 (GraphPad Software, San Diego, CA, USA).

### 2.10. Preparation of VIP116-Loaded Lipodisks for Cellular Assays

A R_eff_ of 0.085 was chosen for all cellular assays based on results from QCM-D experiments. VIP116 (dissolved in DMSO) and lipodisks were incubated in room temperature for 2 h before further diluting in cell media and adding to cells. For XTT viability assays, DMSO was added instead of VIP116 to control wells. For LigandTracer, ^125^I-VIP116 was used instead of VIP116. The maximum concentration of DMSO in the cell assays was 0.12%.

### 2.11. Real-Time Measurement of Cellular Uptake of ^125^I-VIP116 Loaded Lipodisks

The cell-associated uptake of ^125^I-VIP116 when loaded on EGFR-targeted or non-targeting lipodisks was measured using LigandTracer grey (Ridgeview Instruments) at room temperature. Moreover, the uptake of ^125^I-EGF was studied as a reference.

2−3 × 10^5^ HCT-EGFR cells were seeded in a tilted petri dish and incubated for 24–48 h before mounting in the Ligandtracer. A baseline was collected before stepwise adding ^125^I-VIP116-loaded lipodisks to a final total lipid concentration of 2.7, 8 and 24 µM (corresponding EGF concentrations in targeting lipodisks were approximately 3, 9 and 27 nM). The specific activity of ^125^I-VIP116 was 0.5 MBq/nmol in all LigandTracer experiments. For ^125^I-EGF, ligand with concentrations of 1, 3 and 10 nM was used. The background signal was subtracted by measuring the signal on an area with no cells on the same petri dish. Analysis was performed using TraceDrawer version 1.7 (Ridgeview Instruments).

### 2.12. XTT Viability Assays

The cytotoxicity of EGFR-targeted or non-targeting lipodisks with and without VIP116 was measured using XTT viability assays. HCT-116 or HCT-EGFR (1.4 × 10^3^ cells per well) were seeded in 96 well plates and incubated for 48 h. VIP116 was added with concentrations ranging from 1–40 μM and incubated for an additional 72 h. An XTT viability assay (LGC Standards, Wesel, Germany) was performed according to manufacturer’s protocol. In brief, XTT reagent and XTT activation reagent were mixed and added to cell media. Old cell media were removed from cells and the XTT solution was added. After 4 h of incubation, absorbance was measured with a plate reader (BioRad). IC_50_-values were calculated by fitting data to a dose response model (log(inhibitor) vs. response—variable slope).

For XTT viability assays with lipodisks, HCT-EGFR cells were seeded as above and incubated for 72 h. VIP116-loaded lipodisks were prepared as described above. Control solutions consisting of DMSO dissolved in HBS (Ctrl), non-targeting lipodisks with VIP116 (Nontarg + VIP) and targeting lipodisks without VIP116 (Targ + DMSO) were also prepared. Old cell media was removed from wells and lipodisk- or control solutions were added. The total VIP116 concentration in the wells was 40 μM. Plates were incubated for 48 h and an XTT was performed as described above. To compare groups, one-way ANOVA with Bonferroni multiple comparison was performed. 

### 2.13. Statistical Analysis

At least two independent experiments were performed for all assays. Mean and coefficient of variation is presented if not otherwise stated. Statistical analyses, specified in detail in each section above, were done with GraphPad Prism 7 if not otherwise stated where a *p* value ≤ 0.05 was considered significant.

## 3. Results and Discussion

Targeted cancer therapy can improve the therapeutic outcome while minimizing side effects. In the current study, we developed an approach to deliver the p53-activating stapled peptide VIP116 specifically to EGFR-expressing cells by the use of lipodisks.

Measurements using the QCM-D technique confirmed that VIP116 was able to bind to lipodisks with high affinity (Figure 2A,B). The interaction did not measurably affect the lipodisk structure, as suggested by a stable dissipation factor over the course of the experiment (Figure 2A). The values of B_max_ (i.e., maximum effective bound peptide/lipid ratio) and K_D_ obtained by use of a 1:1 binding model (Figure 2B) were 0.11 (± 18%) and 0.973 (± 156%) nM. Based on these findings, peptide to lipid ratio of 0.085 was chosen for the cell assays, to ensure high peptide loading on the disks, while minimizing the amount of free peptide in the solution (99.99 ± 0.015% loaded peptide at the peptide concentration used in cell viability assays).

The large uncertainty in the measured affinity can partly be explained by the slow off-rate (Figure 2B) but it is also possible that the peptide-lipodisk interaction is more complex than assumed by the 1:1 binding model used. It was hypothesized that the peptide bound not only to the disk rim, but also to the flat bilayer part of the disks. Therefore, the interaction between VIP116 and a supported lipid membrane in the gel phase, was investigated (Figure 2C). The R_eff_ for the VIP116-lipid membrane interaction did not increase with concentration and was significantly lower than that of the VIP116-lipodisk interaction across all concentrations. At 100 nM, R_eff_ at equilibrium was 0.01 for the lipid bilayer, i.e., 1/10 of that for the lipodisks. This strongly suggests that the PEG-ylated rim of the lipodisk is responsible for the main part of the interaction with the peptide, an interaction that possibly has multiple components. The details of the interaction are currently being further explored by our group. 

To obtain a suitable model system, the cancer cell line HCT-116 (wild type p53) was transduced with a viral vector containing the EGFR gene. The transduced cell line (referred to as HCT-EGFR) demonstrated a significantly higher uptake of the anti-EGFR antibody ^125^I-cetuximab than HCT-116, with a maximum uptake (B_max_) of 0.34 and 0.01 pmol/10^5^ cells, respectively (Figure 3A). 

For HCT-EGFR, this corresponds to an EGFR density of approximately 2 × 10^6^ receptors per cell. This is in line with EGFR levels reported for the naturally EGFR-expressing cell line A431 [27]. An XTT viability assay confirmed that the sensitivity to VIP116 was not altered by the transduction (*p* = 0.12 for comparison between IC_50_-values, Figure 3B). This validates the relevance and suitability of the model system. In order to verify p53 stabilization by VIP116, HCT-116 cells were treated with VIP116 compound at several concentrations and with Nutlin 3a as a positive control (Figure 3C). p53 levels and Mdm2 exhibited a dose response in correlation with increasing concentrations of the stapled peptide. Additionally, VIP116 as well as the Nutlin-3a positive control also mediated p21 induction. Finally, VIP116-induced p53 transcriptional activity was demonstrated in a *p53* gene reporter assay (Figure 3D), at the same levels of Nutlin-3a positive control. These assays are in line with previous studies on VIP116 and its predecessor PM2, demonstrating that p53 is stabilized by the peptides in wild type p53 cells both in vitro and in vivo [8,9,11,28]. Consequently, these assays verify and validate the relevance and suitability of the model system in the present study.

Cellular binding of lipodisks carrying ^125^I-VIP116 was studied over time on HCT-EGFR cells using LigandTracer (Figure 4A). EGFR-targeted lipodisks demonstrated distinct association and retention phases, while non-targeting lipodisks displayed no measurable binding to the cells. As a comparison, the binding of ^125^I-EGF to HCT-EGFR cells was also characterized with LigandTracer (Figure 4B), demonstrating a similar binding profile as for the EGFR-targeted lipodisks. These results demonstrate that EGF is required for lipodisk-mediated delivery of VIP116 to the HCT-EGFR cells.

To further investigate the cellular uptake of VIP116, the effects of lipodisks on tumor cell viability was studied (Figure 4C). After 48 h incubation with EGFR-targeted lipodisks carrying VIP116 (total VIP116 concentration 40 μM), HCT-EGFR cell viability decreased with 27% compared to controls (*p* < 0.0001). No effects on cell viability was observed after treatment with non-targeting lipodisks carrying VIP116 or “empty” EGFR-targeted lipodisks. This demonstrates an EGF-dependent delivery of VIP116 to the EGFR-expressing cells. Moreover, the results indicate that the peptide is successfully delivered to the cell resulting in a subsequent VIP116-induced therapeutic effect. This is in line with previous studies that have proven that the EGFR-targeted lipodisks are internalized via EGFR [19]. We have previously demonstrated that treatment with the VIP116 predecessor PM2 reduces cell viability in wild type p53 cells, induced by p53-mediated apoptosis, measured by increased cleaved caspase-3 and Noxa activity, as well as senescence, measured by GLB-1 [28]. This is in line with the results in the present study, demonstrating that VIP116 mediates p21 induction, a protein associated with cell cycle arrest and downstream transcriptional target of p53 transcription activity (Figure 3C). This indicates that the VIP116-induced effects on cell viability demonstrated in this study are due to stabilized p53, resulting in subsequent p53-mediated apoptosis, as well as cell cycle arrest and cell senescence via p21. This needs however to be further verified in additional studies.

In conclusion, we demonstrate that lipodisks targeted against EGFR constitute promising vehicles for tumor-specific delivery of the p53-activating stapled peptide VIP116. With this approach, it is possible to increase peptide accumulation in tumor cells while reducing toxicity. The findings reported here merit further studies in relevant disease models using optimized lipodisk formulations and novel receptor targets.

## Figures and Tables

**Figure 1 nanomaterials-10-00783-f001:**
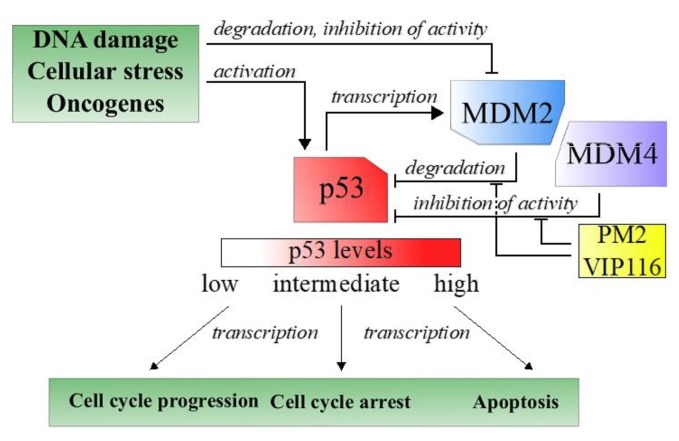
Simplified description of the p53 pathway in response to cellular stress.

**Figure 2 nanomaterials-10-00783-f002:**
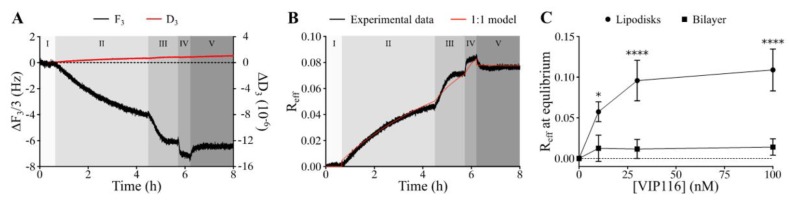
Results from the QCM-D experiments demonstrating the interaction of VIP116 with immobilized lipodisks. (**A**) Raw data displaying changes in the normalized frequency (ΔF/3, black) and dissipation factor (ΔD, red) measured at the 3rd overtone. Different shades and roman numerals represent different stages: baseline (I), association phase upon the addition of 10 (II), 30 (III) and 100 (IV) nM VIP116 and dissociation phase (V) upon rinsing. (**B**) R_eff_ as a function of time as calculated from the QCM-D data (black) and fitted by a 1:1 binding model (red). (**C**) Binding isotherm of VIP116 to lipodisks (circles) and to a supported lipid bilayer (squares), (mean values, SD. N = 3). Solid lines are a guide to the eye.

**Figure 3 nanomaterials-10-00783-f003:**
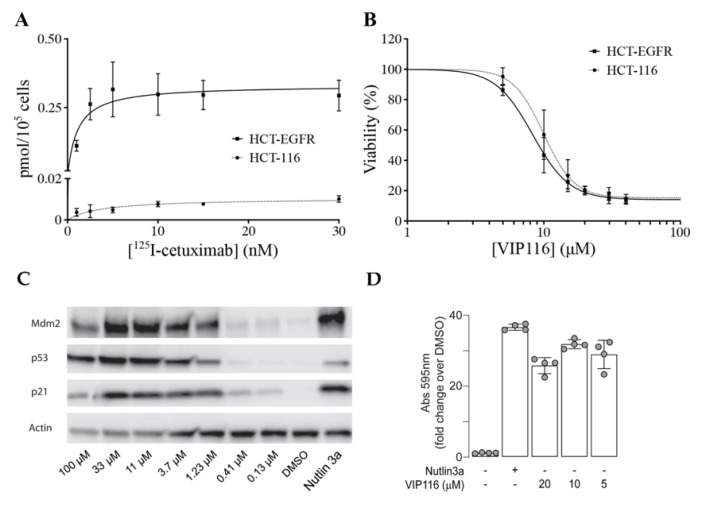
Characterization and validation of cell lines and VIP116. (**A**) Cellular uptake of anti-EGFR ^125^I-cetuximab on HCT-EGFR (squares, solid lines) and HCT-116 (circles, dotted lines) cells, validating the increased EGFR density on HCT-EGFR cells (mean, SD. N = 2). (**B**) XTT viability assay with adjoining dose response curves after 72 h incubation with VIP116, demonstrating retained VIP116 sensitivity for HCT-EGFR cells (mean values, SD. N = 3). (**C**) Western blot analysis of HCT-116 cells, demonstrating VIP116 dose-dependent effects on e.g., p53 levels. Cells were treated with either vehicle control (1% DMSO) or with a 3-fold dilution series from 100 µM to 0.13 µM of VIP116 for 24 h, or with 5 µM of Nutlin 3a as a positive control. VIP116 treatments contained a residual DMSO concentration of 1% *v/v* DMSO. (**D**) *p53* gene reporter assay, demonstrating that VIP116 induces p53 transcriptional activity. ARN8 cells stably expressing the p53 reporter RGC-ΔFos-LacZ were treated with DMSO, Nutlin3a (5 µM) or VIP116 (5, 10, 20 µM) for 18 h. Bar diagram of the mean ± standard deviation of the fold change over DMSO of the absorbance measured at 595 nm in each condition. ANOVA = 0.002. Dots represent technical replicates.

**Figure 4 nanomaterials-10-00783-f004:**
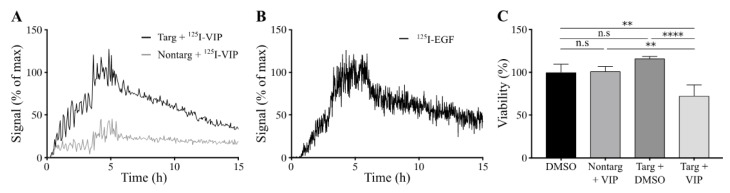
Characterization of lipodisks on HCT-EGFR cells. (**A**) Real-time binding of ^125^I-VIP116 loaded on EGFR-targeted (black) and non-targeting (grey) lipodisks to HCT-EGFR cells. Binding was studied for three consecutive concentrations, followed by a dissociation phase. (**B**) Corresponding real-time binding measurements of ^125^I-EGF on HCT-EGFR cells. (**C**) XTT viability assays on HCT-EGFR cells, demonstrating reduced cell viability in cells treated with EGFR-targeted lipodisks carrying VIP116, whereas no effects were observed from neither non-targeted lipodisks carrying VIP116, nor from EGFR-targeted lipodisks carrying no VIP116 (mean values, SD. N = 3). Nontarg = non-targeting lipodisks. Targ = targeting lipodisks. VIP =VIP116. ** *p* < 0.01, **** *p* < 0.001.

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
