# Peer review of "Tumor-Targeted Delivery of the p53-Activating Peptide VIP116 with PEG-Stabilized Lipodisks"

_nanomaterials, 2020, doi:10.3390/nano10040783_

Round 1

Reviewer 1 Report

The work by Lundsten et al. Reports the use of EGF-decorated lipid bilayer disks for the targeted delivery into EGFR-overexpressing cancer cells of a stapled peptide that disrupts the interaction between the tumor suppressor p53 and its inhibitors MDM2 and MDM4.

The storyline is clear, the experiments are well designed, and the results mostly support the conclusions. Nevertheless, there remain several critical points critical for assessing the therapeutic relevance of the approach used in this paper. Below are specific points to be addressed by the authors:

  1. How do the EGRF protein levels in the used overexpression model compare to those detected in cancer cells naturally expressing high levels of wild type EGFR? (e.g., A431 or A549 cells). Do A549 cells (or similar wt p53/high EGFR cell line) respond to the VP116-EGF-lipodisks treatment?
  2. There seems to be no direct proof that p53 is stabilized/activated by the lipodisks and that it is responsible for the observed impact of the lipodisks on cell viability. Is p53 stabilized in HCT-EGFR cells treated with VIP116-EGF-lipodisks?
  3. The role of p53 in the observed effect of the lipodisks on cancer cell viability should be proved, e.g., by testing the lipodisks activity in HCT-EGFR derived from HCT116 TP53-/- cells.

Minor points:

  1. In their previous work (Yuen et al. 2019, ref. [9]), the authors developed several isomers of the VIP116 peptide. It is not clear which of them was used in the current study.
  2. Line 222: 2% SDS
  3. Line 223: The sensors were
  4. Line 269: was measured

Reviewer 2 Report

The manuscript "Tumor-targeted delivery of the p53-activating peptide 2 VIP116 with PEG-stabilized lipodisks" is an interesting article which deals with the development of lipodisks functionalized with epidermal growth factor and loading p53-activating stapled peptide VIP116. The carriers were designed to the delivery to tumour cells expressing epidermal growth factor. The topic of the articles is very interesting, but the studies are not well reported and discussed. The abstract underlines the promising challenges of the study. The material and Methods section is overly concise and should be separate in different paragraphs. I recommend publication in the Nanomaterials after major revisions.

Materials and Methods are very short and important information on preparation of lipodisks and cell studies are missed.

In the whole manuscript an excessive amount of abbreviations is reported, and they make very difficult the text reading and comprehension. I suggest reducing these and reporting only if the abbreviation is commonly used.

The legends of the Figure 2 and 3, are unclear and must be rewritten to facilitate the understanding of the data reported in the graphs.

The results are not exhaustively presented and discussed.

Round 2

Reviewer 1 Report

The information added by the authors improved the quality of the manuscript. While I still believe that more direct proof of the role of p53 activation in the effect of the lipodisks on cell viability is needed, I agree that the time provided by the editors for the review is short and does not allow for all the necessary experiments to be performed.

Reviewer 2 Report

accepted in the present form